# Immune Cell Ratios Are Higher in Bipolar Affective than Unipolar Depressive Disorder and Modulated by Mood Episode: A Retrospective, Cross-Sectional Study

**DOI:** 10.3390/brainsci13030448

**Published:** 2023-03-05

**Authors:** Anastasia Koureta, Lampros O. Asimakopoulos, Vasilios P. Bozikas, Agorastos Agorastos

**Affiliations:** 1Department of Psychiatry, General Hospital of Katerini, GR 60100 Katerini, Greece; 2Laboratory of Neurophysiology, Faculty of Medicine, School of Health Sciences, Democritus University of Thrace, GR 68100 Alexandroupolis, Greece; 3II. Department of Psychiatry, School of Medicine, Faculty of Medical Sciences, Aristotle University of Thessaloniki, GR 56430 Thessaloniki, Greece

**Keywords:** affective disorders, bipolar disorder, mania, major depressive disorder, neutrophils, platelets, monocytes, biomarkers, immune system, lymphocytes

## Abstract

Immune dysregulation is implicated in the pathophysiology of both bipolar and major depressive disorder, while immune cell ratios (IRCs) have recently been proposed as clinically applicable immune biomarkers. We investigated IRCs differences in affective disorders and their association with current mood episodes and clinical features. This retrospective cohort study analyzed neutrophil–lymphocyte (NLR), monocyte–lymphocyte (MLR), and platelet–lymphocyte (PLR) ratios upon admission in 135 affective disorder in-patients with mania (MA, n = 36), bipolar depression (BiD, n = 38), and unipolar depression (MDD, n = 61). Demographic, clinical, and immune data were extracted from medical records. Monocyte count was significantly higher in BiD compared to MDD (*p* < 0.001). Multivariable regression models suggested higher NLR in MA compared to MDD (*p* = 0.039), higher MLR in both MA and BiD compared to MDD (*p* < 0.001 and *p* = 0.004 respectively), while we found neither group differences in PLR nor an effect of type and duration of hospitalization, current psychotic, or suicidal features and psychiatric history on IRCs. Here, we show that IRCs are elevated in bipolar disorder versus MDD and affected by mood episode, while MLR could be especially valuable in the differential diagnosis between bipolar and unipolar depression. IRCs represent inexpensive, routinely accessible and clinically applicable biomarkers with diagnostic validity in affective disorders that could be easily implemented as illness activity indicators, to better follow the course of illness and eventually predict relapse or treatment response and, thus, guide therapeutic targeting.

## 1. Introduction

The immune system consists of tissues, cell populations, and soluble factors that function to regulate tissue and whole-body homeostasis [1]. Innate and adaptive immune responses and inflammation are host defence mechanisms and an integral part of the human stress response that are crucial to tissue healing, adaptation, and survival [2,3,4]. Immune responses operate within a dynamic range with typical inflammation at one end and graded immune responses of lower magnitude (i.e., para-inflammation) at the other end of the spectrum [5]. The timely regulation of this immune continuum is mediated by intertwined immune, endocrine, and neural regulatory pathways that safeguard against undue tissue and cell damage [6,7].

However, chronic stress activation may compromise the immune functional balance through setting different internal set-points and lead to a dysregulation of the inflammatory status and the cell-mediated immunity, together with other stress-related neurobiological and neuroendocrine responses [8,9,10]. Relatedly, aspects of the immune-inflammatory response, including cell-mediated immunity, are now implicated in stress vulnerability through both peripheral and central mechanisms of action [11,12,13]. Accordingly, adverse psychological conditions such as psychiatric disorders have the potential to function as inducers of a wide range of immune responses, as they chronically influence the human homeostatic stress system [14]. Indeed, both acute and chronic psychosocial stressors in humans have been reliably associated with systemic inflammatory responses [15,16].

Inflammation is additionally involved in oxidative pathways and is a vital mechanism of oxidative stress generation and neurodegeneration in the periphery and the central nervous system, in particular in stress-related disorders [17]. Inflammatory signaling stimulates the activation of oxidative cascades and other the pro-inflammatory cytokines (e.g., interferon-γ, tumor necrosis factor α), chemokines, and prostaglandins, leading to elevated reactive oxygen and nitrogen species [17,18,19]. In addition, inflammation influences the redox equilibrium through the tryptophan-kynurenine pathway [20,21], resulting in elevated glutamate, NMDA receptor overactivation, and excitotoxicity, culminating in reactive nitrogen species generation and lipid peroxidation in neurons, thereby affecting neural function [22].

Thus, the immune system involvement in the pathophysiology of psychiatric and, more specifically, affective disorders has increasingly become a central research focus in the field of immunopsychiatry [23]. Solid research of the past two decades suggests a distinct crosstalk between the central nervous and the peripheral immune system, and has offered mounting evidence of immune dysregulation in affective disorders through several molecular pathways [24]. However, findings are often challenged by several moderating factors, such as phenotype heterogeneity, chronicity and recurrence, current mood state, several methodological issues, but also changes in immune function itself over time (i.e., immunosuppression versus low-grade inflammation) [25]. These factors greatly influence the ability to identify and select clinically applicable, immune-based biomarkers that can help to diagnostically differentiate the specific nature of immune dysfunction across different affective disorders and mood states.

Recently, the interest in simpler, easily accessible, clinically routinely applicable, inexpensive, and reproducible immune biomarkers from peripheral blood has returned to the front. For example, the calculation of routine immune cell count ratios (ICRs), which was first introduced with the monocyte–lymphocyte ratio (MLR) as a new immune biomarker associated with tuberculosis progression back in the 1930s, has been revisited in research about 2 decades ago. Thereby, neutrophil–lymphocyte ratio (NLR) was established by Zahorec et al. in a prospective observational study of patients in ICU setting, suggesting that a simultaneous elevation in the neutrophil count and a decline in the lymphocyte count are associated with most severe critical illnesses [26]. Further research suggested that platelet–lymphocyte ratio (PLR) may represent an additional very sensitive marker of inflammation, as in the absence of absolute thrombocytosis, higher PLR has been interrelated with higher thrombosis and inflammation [27]. Since then, IRCs have been repeatedly proposed as solid prognostic markers for various additional conditions including low-grade inflammation, rheumatic diseases, sepsis, myocarditis, thrombosis, ischemic and cardiovascular events, autoimmune and systemic diseases, cancer incidence, progression, response and outcome, COVID-19 severity and prognosis, critical illness, and overall mortality [26,27,28,29,30,31,32,33,34,35,36,37,38,39,40,41]. Zahorec et al. concluded that ICRs “indicate the balance between innate and adaptive immune responses and are an excellent indicator of inflammation and stress together” [26].

ICRs have been also investigated in various disorders of the central nervous system, such as delirium, Parkinson’s disease, Alzheimer’s disease, multiple sclerosis, and schizophrenia [42,43,44,45,46]. Regarding affective disorders, it was not until 2015, when Demir et al. and Cakir et al. first investigated NLR in MDD and BD respectively, and both reported higher levels of NLR in MDD and BD patients compared to healthy controls [47,48]. Then, 3 years later, a meta-analysis of 11 studies solidly confirmed that NLR and PLR were increased in patients with MDD and BD compared to healthy controls [49]. Interestingly, some studies also pointed out differences in these markers according to the specific affective disorder (i.e., BD vs. MDD) [50] and/or mood episode (manic vs. depressive) [51,52,53,54], suggesting possible fundamental pathophysiological differences across different affective disorder phenotypes. However, all these results call for replication and better investigation of clinical parameters possibly influencing inflammation.

Accordingly, this study aims to investigate differences in NLR, MLR, and PLR in MDD and BD across different mood episodes and their diagnostic validity, as well as their relation to specific clinical features, in order to add to the growing literature, investigate possible modulating factors, and confirm or challenge previous findings.

## 2. Materials and Methods

This retrospective cohort study investigates differences in ICRs in MDD and BD across different mood episodes and their relation to clinical features, in order to add to the growing literature and replicate prior studies. The study was conducted at the Department of Psychiatry of the General Hospital of Katerini, Central Macedonia, Greece. The study protocol was approved by the Ethics Review Committee (ERC) of the Aristotle University of Thessaloniki and the Medical Board of the General Hospital of Katerini.

### 2.1. Patient Sample, Data Extraction, Inclusion, and Exclusion Criteria

Eligible patients (18–65 years of age) that were diagnosed with BD or MDD, experiencing a manic or depressive mood episode of at least moderate severity (bipolar or unipolar) and who were admitted to the Psychiatric Unit throughout a period of seven years (January 2013–December 2019) were included. Medical history was collected from the medical records from two blinded psychiatrists and included only if confirmed by both. The clinical evaluations and diagnoses were carried out by experienced psychiatrists through several in-person psychiatric interviews according to International Classification of Diseases and Related Health Problems, tenth revision (ICD-10) criteria. The exclusion criteria were (a) the presence or self-reported history of any chronic or acute psychotic disorder (e.g., schizophrenia, schizoaffective disorder, etc.); (b) mixed mood episodes due to the small number (n = 5); (c) acute or chronic physical comorbidities at a current dysregulated or exacerbated state according to consulting internal medicine physicians that could affect immune status (e.g., infection, diabetes mellitus, autoimmune disease, cardiovascular diseases, dyslipidaemia, cancer, osteoporosis, asthma, chronic respiratory disease, trauma); (d) co-administration of drugs such as non-steroid anti-inflammatory drugs, steroids, antibiotics, chronic non-opioid, and opioid pain relief medications); (e) present or former substance use disorder; or (f) admission due to social reasons. Additional data were extracted manually from hospital medical records and recorded, archived, and stored electronically in an anonymized encrypted form. As the study was a retrospective cohort study, according to international ethics protocols, written consent of patients was not required.

### 2.2. Blood Samples

The results from routine blood samples upon admission were included in this study. Blood sampling was conducted by trained phlebotomists in the first morning (8 a.m.–10 a.m.) following admission of each patient to the clinic. Blood samples were then immediately transported to the Laboratory Department of the Hospital, where they were assessed and analyzed according to standardized hospital laboratory protocols.

### 2.3. Statistical Analyses

The sample was divided into three diagnostic groups according to mood episode upon admission (manic, MA; bipolar affective depressive episode, BiD; and unipolar major depressive episode, MDE). NLR, MLR, and PLR were calculated as the ratio of each cell type. Normality was assessed by using the Shapiro-Wilk test. The equality of variances was assessed by using the Levene’s test. The level of statistical significance was set at *p*-value < 0.05. All statistical analysis was conducted by using RStudio version 4.0.3.

#### 2.3.1. Univariate Group Differences

Differences between the groups were assessed through parametric (one-way analysis of variance, ANOVA) and non-parametric tests (Kruskal–Wallis test) according to the distribution of the assessed variables. Post hoc analyses were conducted either by the pairwise t-test or the Wilcoxon–Mann–Whitney test. Adjusted *p*-values were calculated by using Bonferroni correction.

#### 2.3.2. Linear Regression Analyses

For each calculated ICR, a linear regression model was used considering each ICR as the dependent value, the three diagnostic groups as fixed factors, and age, sex, suicide attempt, current psychotic features, and medical history of BD or MDD as confounders.

#### 2.3.3. Multivariable Analyses

In the multivariable analysis, the candidate explanatory variables for adjustment were selected by using the backward elimination method (for detailed description cf. Table 3, legend). Backward elimination starts with the full model including all candidate explanatory variables and sequentially removes the least contributing variables, resulting in the best performing model. The final model was selected according to Akaike information criterion (AIC). AIC is a measure of the goodness of fit for statistical models and is used in model selection. The model with the lowest AIC was selected as the best performing model. In all multivariable analyses, mood episode was used as fixed factor (even if the AIC was lower by excluding the mood episode, this variable was included in the final model, based on the research question).

## 3. Results

### 3.1. Basic Characteristics of the Sample

From the 503 total admissions between January 2013 and December 2019, 135 met the inclusion criteria. The final sample had a mean (SD) age of 47.4 (13.6) years and was comprised of 30% (n = 40) males. Of the total sample, 38 admissions (28.1%) were due to a BiD, 36 (26.7%) due to MA, and 61 (45.2%) due to MDE. Further clinical characteristics of the total sample are summarized in Table 1.

### 3.2. Group Differences

Basic laboratory values did not differ between the groups, with the exception of monocyte count. Immune values and comparisons for all three groups are presented in Table 2. After implementing post hoc analysis and Bonferroni correction, the BiD group showed a significantly higher monocyte count compared to the MDE group (*p* < 0.001), while there was no difference between MA and BiD or MDE. With respect to ICRs, PLR showed no statistically relevant differences between the groups (*p* = 0.785; cf. Table 2, Figure 1). Regarding NLR, MA patients showed higher values compared to BiD or MDE patients, however only at a statistical trend level (*p* = 0.056; cf. Table 2, Figure 1). MLR, however, differed significantly between the groups (*p* < 0.001, cf. Table 2, Figure 1). Post hoc analysis after Bonferroni correction showed that MLR was significantly higher in MA compared to MDE (*p* = 0.005), but not to BiD patients (*p* = 0.916). MLR also significantly differed between BiD and MDE, with BiD patients showing higher values (*p* = 0.002).

### 3.3. Univariate Analyses

#### 3.3.1. Neutrophil–Lymphocyte Ratio (NLR)

In the univariable analysis, there was no association between NLR and age, sex, suicide attempt, and psychotic features upon admission. NLR was significantly lower in patients with bipolar depressive and with unipolar depressive episode compared to patients with mania. There was no statistically relevant difference between bipolar and unipolar depression.

#### 3.3.2. Platelet–Lymphocyte Ratio (PLR)

In the univariable analysis, there was no association between PLR and sex, current mood episode, suicide attempt, and psychotic symptomatology upon admission, and history of BD or MDD. Between PLR and age, there was an expected decrease of 0.84 units of PLR for every 1-year increase in age (95% CI: −1.53 to −1.49, *p* = 0.018).

#### 3.3.3. Monocyte–Lymphocyte Ratio (MLR)

In the univariable analysis, there was no association between MLR and age, suicide attempt, and psychotic symptomatology upon admission, and history of BD or MDD. MLR was significantly higher in men compared to women and lower in patients with unipolar depressive episode compared to patients with mania and to patients with bipolar depressive episode, while there was no association between bipolar depressive episode and mania.

### 3.4. Multivariate Regression Models

#### 3.4.1. Neutrophil–Lymphocyte Ratio (NLR)

In the multivariable analysis, after exclusion of variables age, sex, suicide attempt upon admission, and history of BD or MDD by using the backward selection method, NLR remained significantly lower in patients with MDE, but not with BiD, compared to patients with MA, adjusted for psychotic symptomatology upon admission (95% CI: −0.93 to −0.02, *p* = 0.039) (cf. Table 3).

**Table 3 brainsci-13-00448-t003:** Univariable analyses and multivariate regression models for NLR, PLR, MLR, and explanatory variables.

	Univariable Analysis			Multivariable Analysis ^a^		
Variable	Unadj. *β*	95% CI	* p *	Adj. *β*	95% CI	* p *
** NLR **						
**Age** (years)	−0.002	(−0.015, 0.011)	0.787	-	-	-
**Sex** (Male/Female)	0.207	(−0.187, 0.601)	0.301	-	-	-
**Episode**						
BiD/MA	**−0.487**	**(−0.968, −0.007)**	**0.047**	−0.421	(−0.932, −0.089)	0.105
MDE/MA	**−0.528**	**(−0.949, −0.106)**	**0.015**	**−0.475**	**(−0.926, −0.024)**	**0.039**
MDE/BiD	−0.04	(−0.466, 0.385)	0.852	−0.054	(−0.486, 0.379)	0.807
**Suicide attempt**						
Yes/No	−0.197	(−0.755, 0.362)	0.487	-	-	-
**Psychotic features**						
Present/Absent	0.285	(−0.083, 0.653)	0.128	0.134	(−0.257, 0.525)	0.498
**History of BD/MDD**						
Yes/No	0.32	(−0.357, 0.997)	0.351	-	-	-
** PLR **						
**Age** (years)	**−0.84**	**(−1.53, −0.148)**	**0.018**	**−0.788**	**(−1.5, −0.082)**	**0.029**
**Sex** (Male/Female)	14.4	(−6.95, 35.8)	0.184	-	-	-
**Episode**						
BiD/MA	−0.019	(−26.9, 26.8)	0.999	−3.88	(−30.9, 23.1)	0.777
MDE/MA	−6.98	(−30.5, 16.6)	0.558	−10.10	(−33.8, 13.6)	0.401
MDE/BiD	−6.96	(−30.8, 16.8)	0.563	−6.20	(−29.9, 17.5)	0.605
**Suicide attempt**						
Yes/No	23.3	(−6.88, 53.5)	0.129	21.8	(−8.57, 52.2)	0.158
**Psychotic features**						
Present/Absent	3.42	(−16.8, 23.7)	0.738	-	-	-
**History of BD/MDD**						
Yes/No	20.8	(−16.0, 57.7)	0.265	-	-	-
** MLR **						
**Age** (years)	0	(−0.001, 0.001)	0.897	-	-	-
**Sex** (Male/Female)	**0.042**	**(0.005 0.079)**	**0.027**	-	-	-
**Episode**						
BiD/MA	−0.011	(−0.055, 0.034)	0.632	−0.027	(−0.073, 0.020)	0.258
MDE/MA	**−0.073**	**(−0.112, −0.034)**	**<0.001**	**−0.084**	**(−0.125, −0.043)**	**<0.001**
MDE/BiD	**−0.062**	**(−0.101, −0.023)**	**0.002**	**−0.057**	**(−0.096, −0.019)**	**0.004**
**Suicide attempt**						
Yes/No	−0.004	(−0.057, 0.049)	0.876	-	-	-
**Psychotic features**						
Present/Absent	−0.004	(−0.04, 0.031)	0.804	−0.027	(−0.062, 0.009)	0.14
**History of BD/MDD**						
Yes/No	0.043	(−0.022, 0.107)	0.193	-	-	-

For each calculated immune cell ratio, a linear regression model was used considering the ratio as the dependent value, and the three diagnostic groups were used as fixed factors, while age, sex, suicide attempt, current psychotic features, and medical history of BD or MDD were used as confounders. In the multivariable analysis, the candidate explanatory variables (age, sex, suicide attempt, psychotic features, and medical history of BD or MDD) for adjustment were selected by using the backward elimination method. The final model was selected according to Akaike information criterion (AIC). The model with the lowest AIC was selected as the best performing model. In all multivariable analyses, mood episode was used as a fixed factor (even if the AIC was lower by excluding the mood episode, this variable was included in the final model, based on the research question). For NLR, the variable psychotic features during the current episode was used as a confounder, while variables age, sex, suicide attempt upon admission, and history of BD or MDD were excluded, as indicated by AIC. For PLR, the variables age and suicide attempt upon admission were used as confounders, while the variables sex, psychotic symptomatology upon admission, and history of BD or MDD were excluded. For MLR, the variable psychotic features during the current episode was used as a confounder, while variables age, sex, suicide attempt upon admission, and history of BD or MDD were excluded. In the final model, a check for linearity, homoscedasticity, and normality of the residuals was carried out, while multicollinearity was estimated by the variance inflation factor (VIF). A VIF lower than 5 indicated no multicollinearity between the explanatory variables. *β*: coefficient of the explanatory variable, CI: Confidence interval, BiD: Bipolar depressive episode, MDE: Unipolar major depressive episode, MA: Manic episode; NLR: Neutrophil–Lymphocyte Ratio; PLR: Platelet–Lymphocyte ratio; MLR: Monocyte–Lymphocyte ratio. ^a^ Backward selection method. Significant results are bolded.

#### 3.4.2. Platelet–Lymphocyte Ratio (PLR)

In the multivariable analysis, after exclusion of the variables sex, psychotic symptomatology upon admission, and history of BD or MDD by using the backward selection method, the negative association between PLR and age remained significant adjusted for current mood episode and suicide attempt upon admission (95% CI: −1.50 to −0.08, *p* = 0.029), while there was no association between PLR and other variables.

#### 3.4.3. Monocyte–Lymphocyte Ratio (MLR)

In the multivariable analysis, after exclusion of the variables age, sex, suicide attempt upon admission, and history of BD or MDD by using backward selection method, MLR remained significantly lower in patients with MDE compared to patients with MA, adjusted for psychotic symptoms upon admission (95% CI: −0.12 to −0.04, *p* < 0.001), and to patients with BiD adjusted for psychotic symptoms upon admission (95% CI: −0.1 to −0.02, *p* = 0.004).

### 3.5. Synopsis of Main Findings

Taken together, after implementing multivariable linear regression models, our study reported higher NLR in MA compared to MDE, and higher MLR in MA and BiD compared to MDE, while there was no association between PLR and different mood episodes. In addition, our results suggested no effect of type and duration of hospitalization, current psychotic or suicidal features, and psychiatric history on ICRs.

## 4. Discussion

Until recently, most studies assessing potential differences in immune status in BD and MDD have focused mostly on cytokines and other inflammatory markers (e.g., C-reactive protein, CRP). However, only a few studies have investigated ICRs, specifically in BD compared to MDD. This retrospective cohort study aimed to investigate differences in ICRs in patients with BD and MDD across different mood episodes (MA, BiD, MDE), in order to add to the growing literature and confirm or challenge previous findings. In particular, our study aimed to replicate and extend previously reported findings [50,51] that compared ICRs in mania, bipolar depression, and unipolar depression. Our study additionally controlled for the first time, the modulating role of important clinical confounders such as type and duration of hospitalization, current psychotic or suicidal features, and personal history of BD or MDD on ICRs across different mood episodes. All these variables have been considered to affect immune response in patients with mood disorders to some extent.

Our main results suggest higher NLR in MA compared to MDE, and higher MLR in MA and BiD compared to MDE, while we found no association between PLR and different mood episodes. In addition, our results showed no effect of type and duration of hospitalization, current psychotic or suicidal features and psychiatric history on ICRs. Taken together, these findings suggest a more severe inflammatory dysregulation during mania than in depression and that potential pathophysiological differences between bipolar and unipolar depression may exist, with a shift towards higher inflammation in bipolar depression and, thus, bipolar disorder in general.

These results are very interesting when compared to previous studies. Mazza et al., who first aimed to associate ICRs with different mood episodes in BD and MDD, demonstrated higher NLR and MLR in mania compared to both bipolar and unipolar depression, while they did not report differences between bipolar and unipolar depression [50]. Similarly, in a very recent cohort study, Dionisie et al. reported significant differences concerning NLR, MLR, and PLR between mania and unipolar depression, while NLR was also higher in bipolar depression compared to unipolar depression [51]. These results were comparable to a recent, large-scale, retrospective study by Wei et al. [55], which reported elevated NLR in MDD patients in comparison to healthy controls, higher NLR and MLR and lower PLR in BD patients in comparison to MDD, and elevated NLR and MLR in BD patients with manic episode compared to BD patients with depression or MDD patients. Other studies also reported similar results. For example, Inanli et al. reported higher NLR and MLR in manic state compared to depressive state in BD patients, while depressed patients showed higher ratios compared to healthy controls [53]. Furthermore, Fusar-Poli et al. found higher immune ratios in mania compared to depression in individuals with BD [52], while Dadouli et al. reported higher immune ratios in patients with BD than in controls when experiencing a depressive or manic episode, while MLR was higher only in BD patients with a manic episode [56].

Our study is somewhat consistent with these prior studies, as we not only found higher NLR and MLR in mania compared to unipolar depression, but also higher MLR in bipolar compared to unipolar depression. Most interestingly, MLR was the only immune ratio that was clearly associated with BD independent of mood episode, highlighting possible higher immune activation in bipolar depression expressed in peripheral immune cells. Our findings, together with the previous studies, suggest that BD shows a greater immune dysregulation with higher ICRs compared to MDD and that immune state in BD could be modulated by mood episode, with manic episodes consistently showing higher ICRs. These results also support a fundamental difference in the pathophysiology of BD and MDD independently of current mood state. On the other hand, our study does not replicate some of the previous findings, which can be partly explained by the phenotypical plurality of affective stages in addition to diverging course and recurrence of illness, as a chronic or frequently recurrent state may alter the neurobiology of the disorder over the years.

Finally, the secondary results of our study indicated decreasing PLR with age, and higher MLR in men compared to women. These sex-related results are in agreement with a study by Moosazadeh et al. that showed similar sex differences in MLR [57], but were not found elsewhere [56]. On the other hand, the PLR decreased with age, which is not consistent with previously reported results [57]. Regarding suicidality, previously studies reported higher NLR and PLR in depressed patients with suicidal ideation and behaviour [58], which our study could not confirm.

Finally, some limitations merit discussion. First, due to lacking information in the majority of patients’ medical records, our study did not take BMI, smoking, medication prior to submission, and clinical rating scales into consideration as possible confounders. Second, other immune and inflammatory biomarkers (i.e., cytokines, CRP) were not evaluated, as they were not part of a routine admission laboratory protocol. Furthermore, patients were not compared to a matched healthy control group in order to investigate differences in ICRs. On the other hand, our study was one of the few comparing both manic and depressive BD episodes to MDD, and also investigating the effects of type and duration of hospitalization, current psychotic, or suicidal features and psychiatric history on immune cell ratios. In addition, our sample size was comparable with several prior studies in the field [47,48,50,56].

Nevertheless, further large-scale, prospective cohort studies are needed in order to confirm and expand these results and test them for sensitivity and specificity across mood episodes and diagnosis, in order to integrate ICRs into every day clinical psychiatric practice. Immune biomarkers and ICRs could be used in three contexts, in prediction (i.e., relapse prevention indicator), staging (i.e., illness activity indicator), and treatment response (i.e., response to treatment indicator) of affective disorders [46,59,60]. Last but not least, a potential combination of several immune biomarkers into a systematic index to assess dynamic immune and inflammatory states as in prior studies [61] could be of particular interest in future studies.

Our results support the use of ICRs in the diagnosis and proper classification of affective disorders across their different episodes. Especially, the differential diagnosis between a first bipolar or unipolar depressive episode is critical, as these two states might not differ phenotypically, but require a completely different therapeutic approach. ICRs could, therefore, be of major diagnostic importance as novel, clinically applicable, cheap, and easily accessible immune biomarkers that could guide clinicians towards a better diagnosis and treatment in affective disorders. Additionally, the broader use of immune biomarkers in clinical research might also boost further basic research targeting novel neuroimmune pathways of interest and the development of new therapeutic targets in mood disorders. So far, systematic reviews and meta-analyses support the potential therapeutic effects of anti-inflammatory drugs as adjunctive agents to both bipolar disorder and major depression [62,63].

Taken together, this study presented important data concerning the association of immune state, as measured by changes in ICRs, and current mood episode in affective disorders. Our results support higher NLR and MLR in mania and higher MLR in bipolar depression compared to unipolar depression, while type and duration of hospitalization, current psychotic or suicidal features and history of BD or MDD did not affect our results. Our findings are in line with the hypothesis that ICRs are higher in BD than in MDD and that immune state in BD is modulated by mood episode. MLR could, particularly, be especially valuable in the differential diagnosis between bipolar and unipolar depression, in particular in first-episode depressed patients with no psychiatric history.

## 5. Conclusions

Simple, inexpensive, and clinically applicable immune biomarkers, such as ICRs, could be easily implemented as illness activity indicators, in order to better support diagnosis, objectively follow course, and eventually predict and confirm relapse or treatment response in affective disorders. Such a step might not only be relevant for pathophysiological and diagnostic purposes, but also in light of therapeutic targeting in affective disorders, as several systematic reviews and meta-analyses have already supported the potential benefit of anti-inflammatory drugs as adjunctive agents in specific affective phenotypes [62,63,64].

## Figures and Tables

**Figure 1 brainsci-13-00448-f001:**
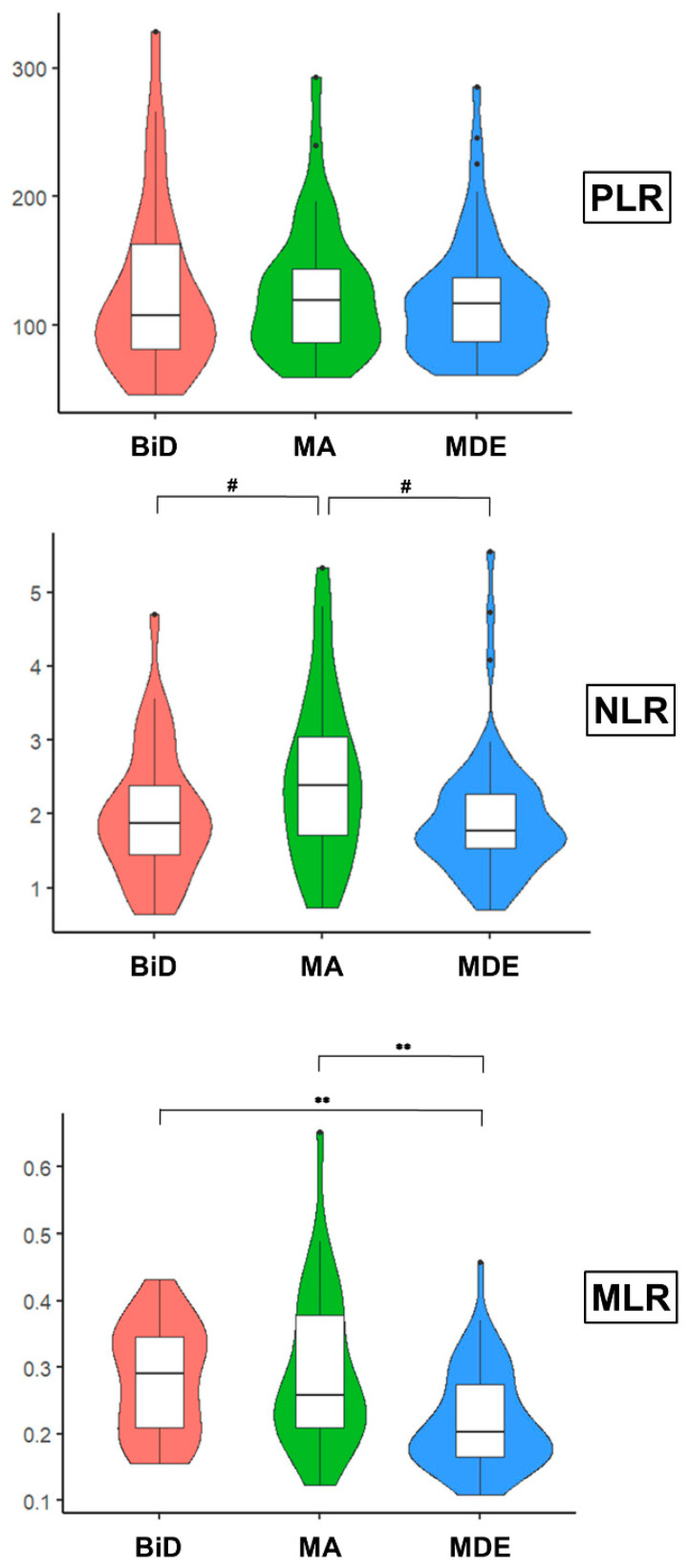
Violin plots and boxplots of PLR, NLR, and MLR in patients with MA, BiD, and MDE. Violin plots and boxplots showing PLR, NLR, and MLR medians (interquartile range, IQR) in patients with MA, BiD and MDE accordingly. BiD: Bipolar depressive episode, MDE: Unipolar major depressive episode, MA: Manic episode; NLR: Neutrophil–Lymphocyte Ratio; PLR: Platelet–Lymphocyte ratio; MLR: Monocyte –Lymphocyte ratio. ** *p* < 0.01, ^#^ 0.05 < *p* < 0.1.

**Table 1 brainsci-13-00448-t001:** Basic demographic and clinical characteristics of the total sample (N = 135).

Characteristic	Value/Mean/Median (%/SD/IQR)
Gender	
Male	40 (30%)
Female	95 (70%)
Age (years)	47.4 (13.6)
Mood Episode	
Bipolar Depression	38 (28%)
Mania	36 (27%)
Unipolar Major Depression	61 (45%)
Duration of hospitalization (days)	17 (20)
Type of hospitalization	
Voluntary	90 (67%)
Involuntary	45 (33%)
Suicide attempt upon admission	
Yes	14 (10%)
No	121 (90%)
Psychotic features	
Present	49 (37%)
Absent	86 (63%)
History of BD or MDD	
Yes	12 (9%)
No	123 (91%)

SD: standard deviation; IQR: interquartile range.

**Table 2 brainsci-13-00448-t002:** Differences in blood laboratory values and immune cell ratios across diagnostic groups.

Variable	Bipolar Depressive Episode (BiD)	Manic Episode (MA)	Unipolar Depressive Episode (MDE)	* p *	* post hoc *
WBCs (10^3^/μL)	7.63 (2.09)	7.98 (3.50)	7.13 (2.74)	0.370 ^1^	
Neutrophils (10^3^/μL)	4.49 (2.19)	5.12 (3.55)	4.27 (1.61)	0.182 ^1^	
Lymphocytes (10^3^/μL)	2.33 (0.94)	2.35 (0.67)	2.27 (0.83)	0.455 ^2^	
Monocytes (10^3^/μL)	0.70 (0.26)	0.54 (0.29)	0.48 (0.16)	**<0.001 ^1,^****	**BiD > MDE**
Eosinophils (10^3^/μL)	0.19 (0.14)	0.13 (0.11)	0.12 (0.14)	0.095 ^1^	
Basophils (10^3^/μL)	0.02 (0.02)	0.02 (0.02)	0.02 (0.02)	0.154 ^1^	
RBCs (10^6^/μL)	4.60 (0.81)	4.72 (0.55)	4.60 (0.42)	0.323 ^2^	
Haemoglobin (g/dL)	13.9 (1.45)	13.8 (1.60)	13.5 (2.20)	0.457 ^2^	
Haematocrit (%)	40.8 (4.15)	40.8 (5.70)	40.4 (5.20)	0.434 ^2^	
PLTs (10^3^/μL)	241 (128)	240 (112)	255 (119)	0.988 ^1^	
NLR	1.88 (0.94)	2.39 (1.34)	1.78 (0.73)	0.056 ^1,#^	
PLR	107 (81.3)	119 (57.8)	117 (49.9)	0.785 ^1^	
MLR	0.29 (0.14)	0.26 (0.17)	0.20 (0.11)	**<0.001 ^1,^****	**MA > MDE ** **BiD > MDE**

The results are presented as medians (IQR). *p*-values of statistical significance and trend level are bolded. WBCs: White blood cells; RBCs: Red blood cells; PLTs: Platelets; NLR: Neutrophil–Lymphocyte Ratio; PLR: Platelet–Lymphocyte ratio; MLR: Monocyte–Lymphocyte ratio. ^1^ Kruskal–Wallis test. ^2^ ANOVA test. ** *p* < 0.01. ^#^ 0.05 < *p* < 0.1. Significant results are bolded.

## Data Availability

The data presented in this study are available on request from the corresponding author. The data are not publicly available due to ethics protocol.

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
