# Peer review of "Immune Cell Ratios Are Higher in Bipolar Affective than Unipolar Depressive Disorder and Modulated by Mood Episode: A Retrospective, Cross-Sectional Study"

_brainsci, 2023, doi:10.3390/brainsci13030448_

Round 1

Reviewer 1 Report

The Authors proposed an interesting study regarding the influencing role of blood inflammatory biomarkers in affective disorders. 

The study is in line with an important research topic in which chronic inflammation is becoming a fundamental risk factor for many disorders. 

In this light, the use of NLR, PLR, MLR and also SII are becoming more and more widely accepted as easy and reliable surrogates of immune disequilibrium. 

The paper is well-written and clearly presented. 

I have a major concern about the exclusion criteria that were used in the study. The Authors decided to eliminate several variables "acute or chronic physical comorbidities that could affect immune status (e.g., infection, diabetes mellitus, autoimmune 93 disease, cardiovascular diseases, dyslipidemia, cancer, osteoporosis, asthma, chronic respiratory disease, trauma)"

I understand that this choice was made to try to reduce as much as possible the number of confounding factors. I agree to exclude events such as acute cardio- and cerebrovascular, traumas or acute infection but I don't fully agree with the decision of excluding chronic condition that are part of patient baseline status and especially inflammatory status. Please comment. 

In addition, I think that the introduction section could be somewhat implemented in terms of the use of these biomarkers in others medical and surgical field. Very few references are reported regarding cardiovascular condition that are very common in everyday clinical practice [doi: 10.3390/biomedicines10092218. ; doi: 10.5681/jcvtr.2014.007. 

Moreover, these biomarkers heve been also linked to acute fluctuating psychic condition as delirium and post-operative delirium in different medical and surgical settings. [ doi: 10.1186/s12957-022-02793-x.

Lastly, conclusion section is too long, please move something in the discussion section. Conclusion must be brief and coincise.  

Author Response

Comment #1: The Authors proposed an interesting study regarding the influencing role of blood inflammatory biomarkers in affective disorders. The study is in line with an important research topic in which chronic inflammation is becoming a fundamental risk factor for many disorders. In this light, the use of NLR, PLR, MLR and also SII are becoming more and more widely accepted as easy and reliable surrogates of immune disequilibrium. The paper is well-written and clearly presented.

Response: We would like to thank the Reviewer for these positive comments!

Comment #2: I have a major concern about the exclusion criteria that were used in the study. The Authors decided to eliminate several variables "acute or chronic physical comorbidities that could affect immune status (e.g., infection, diabetes 
mellitus, autoimmune disease, cardiovascular diseases, dyslipidemia, cancer, 
osteoporosis, asthma, chronic respiratory disease, trauma)". I understand that this choice was made to try to reduce as much as possible the number of confounding factors. I agree to exclude events such as acute cardio- and cerebrovascular, traumas or acute infection but I don't fully agree with the decision of excluding chronic condition that are part of patient baseline status and especially inflammatory status. Please comment.

Response: We would like to thank the Reviewer for raising this important point. 
We acknowledge that we have not described properly this exclusion criterion in our methods, as indeed only chronic conditions in a current dysregulated or exacerbated state according to our consulting internal medicine colleagues have been actually excluded in our study. This is now properly described in the methods section.

Comment #3: The introduction section could be somewhat implemented in terms of the use of these biomarkers in others medical and surgical field. Very few references are reported regarding cardiovascular condition that are very common in everyday clinical practice [doi: 10.3390/biomedicines10092218. ; doi: 
10.5681/jcvtr.2014.007. Moreover, these biomarkers have been also linked to acute fluctuating psychic condition as delirium and post-operative delirium in different medical and surgical settings. [ doi: 10.1186/s12957-022-02793-x.

Response: We acknowledge the Reviewer’s comment and have added some 
additional text including several references (incl. the suggested ones) with respect to the use of immune ratios in other medical fields, as well as additional references with respect to other psychiatric disorders.

Comment #4: Lastly, conclusion section is too long, please move something in the discussion section. Conclusion must be brief and concise.

Response: We acknowledge the Reviewer’s comment and have followed this 
suggestion by moving some text parts into the discussion section. 

Reviewer 2 Report

19 February 2023 

Manuscript ID: brainsci-2253706

Type: Article

Title: ‘Immune cell ratios as diagnostic biomarkers in affective disorders: A retrospective, cross-sectional study’ by Koureta A et al., submitted to Brain Sciences 

Dear Authors,

The present research article, entitled ‘Immune cell ratios as diagnostic biomarkers in affective disorders: A retrospective, cross-sectional study’ by Koureta and colleagues conducted a retrospective study to investigate the immune cell ratios of patients with mania, bipolar depression, and unipolar depression.

The main strength of this manuscript is that it addresses an interesting and timely question, reporting that immune cell ratios are higher in patients with bipolar depression than those with unipolar depression.

In general, I think the idea of this original article is interesting and the authors’ fascinating observations on this timely topic may be of interest to the readership of Brain Sciences. However, some comments as well as some crucial evidence should be included to support the author’s argumentation to improve its adequacy, its readability, and thus the quality of the manuscript, prior to publication. My overall opinion is that  the author should carefully considered my suggestions below during the peer-review session.

Please consider the following comments:

1.      Title: This is the most important section of the manuscript. Please present a concise and self-explanatory title stating the most important message of this study.

2.      Abstract: I would like the authors to make as much effort for this section as for the rest of the manuscript. Please abridge the abstract to 200 words according to the guidelines of the journal (https://www.mdpi.com/journal/brainsci/instructions), proportionally presenting the background, the methods, the results, and the conclusion without subheadings.

The background should include the general background (one to two sentences), the specific background (two to three sentences), and current issue addressed to this study (one sentence), leading to the objectives. In this subsection I would like the authors to lay out basic information, problem statement, and the authors’ motivation to break off.

The methods should clarify the authors’ approach such as study design and variables to solve the problem and/or to make progress on the problem. In this abstract a group of healthy control is not included. So, please justify the study design.

The results section must state the results in numbers and clarify their statistical significance. So, please avoid using ‘higher’. Are the results statistically significant? This subsection should close with one to two sentences which put the results into a more general context.

The conclusion should include one sentence describing the main result using such words like “Here we show”. The conclusion should write the potential and the advance this study has provided in the field and finally a broader perspective (two to three sentences) readily comprehensible to a scientist in any discipline.

3.      Keywords: Please focus on choosing ten keywords from Medical Subject Headings (MeSH) (https://meshb.nlm.nih.gov/). I recommend adding ‘mania’ or equivalent as a keyword and use as many as possible in the title and in the first two sentences of the abstract

4.      A graphical abstract that will visually summarize the main findings of the manuscript is highly recommended.

5.      Introduction: The author needs to fully expand this section with up to 1000 words, as it is short of information on the main constructs of this study, which should be understood to a reader in any discipline and make persuasive enough to put forward the main purpose of current research the author has conducted and the specific purpose the authors has intended by this study. I would like to encourage the author to present the introduction starting with the general background, proceeding to the specific background, and finally the current issue addressed to this study, leading to the objectives. Those main structures should be organized in a logical and cohesive manner. In this regard, I would suggest providing a general overview of pathogenesis and biochemical hallmarks of inflammation and mental illness, for example Mitochondrial impairment as a common motif in neuropsychiatric presentation (https://doi.org/10.3390/cells11162607) and ‘Immune influencers in action in inflammation and kynurenines. Moreover, I would recommend adding more information on neural substrates of cognitive dysfunction presented in mental illnesses as in ‘Functional interplay between central and autonomic nervous system’. This information may provide a better understanding of prefrontal cortex’s key role and how its disrupted function may contribute to irregular behavioral responses and therefore to the development of many cognitive dysfunctions that are common in encephalopathy (https://doi.org/10.3389/fnbeh.2022.998714).

6.      Methods: I recommend including short introductory paragraphs regarding the study design and citing more references to ensure the reliability and the integrity of evidence in the study design the authors have built and the methodology the authors applied to this study. Also, I recommend using more paragraphs or subheadings according to the type of analysis, as in the results.

7.      Results: I would like the authors to organize this section according to the methods section, so that a reader could easily follow the results. I also expect the authors to close this section with a paragraph which put the results into a more general context.

8.      Discussion: The authors need to fully expand this section with up to 1500 words. Some essential elements for discussion are described, but I recommend reorganizing clarifying the following elements. Starting with the summary of the previous section (Results), the authors need to develop discussion on the potential of this study complementing as the extension of the previous work, the implication of the findings of this study, how this study could facilitate future research, the ultimate goal, the challenge, the knowledge and the technology necessary to achieve this goal, the statement about this field in general, and finally the importance of this line of research. It is particularly important to present its limit and its merit, and its potential translation of this study to clinical application.

9.      Conclusion: I think that this section would benefit from a single paragraph (which should be shorter than the current conclusion) presenting some thoughtful as well as in-depth considerations by the author as an expert to convey the take-home message, as it is very descriptive but not enough theoretical as a conclusion should be. The author should make his effort to explain the theoretical implication as well as the translational application of their research.

10.  References: Please follow the guidelines of the journal (https://www.mdpi.com/journal/brainsci/instructions) and cite more references to support introductory information and arguments for the discussion section. Typically, the original study like this needs 60-70 references.

Overall, the manuscript contains three figures, three tables, and 26 references. I believe that the manuscript may carry important value in studying the immune cell ratios of patients with mood disorders. I hope the manuscript will meet the high standard of the journal for publication following the peer-review session. I am available for a new round of revision of this paper.

Best regards,

Reviewer

Author Response

Comment #1: The main strength of this manuscript is that it addresses an interesting and timely question, reporting that immune cell ratios are higher in patients with bipolar depression than those with unipolar depression. In general, I think the idea of this original article is interesting and the authors’ fascinating observations on this timely topic may be of interest to the readership of Brain Sciences.

Response: We would like to thank the Reviewer for these positive comments!

Comment #2: Title: This is the most important section of the manuscript. Please 
present a concise and self-explanatory title stating the most important message of this study.

Response: We acknowledge this comment and have revised our title 
accordingly. We also kindly refer to Comment #2 of Reviewer 3.

Comment #3: Abstract: I would like the authors to make as much effort for this section as for the rest of the manuscript. Please abridge the abstract to 200 words according to the guidelines of the journal (https://www.mdpi.com/journal/brainsci/instructions), proportionally presenting the background, the methods, the results, and the conclusion without subheadings. The background should include the general background (one to 
two sentences), the specific background (two to three sentences), and current issue addressed to this study (one sentence), leading to the objectives. In this subsection I would like the authors to lay out basic information, problem statement, and the authors’ motivation to break off. The methods should clarify the authors’ approach such as study design and variables to solve the problem and/or to make progress on the problem. In this abstract a group of healthy control is not included. So, please justify the study design. The results section must state the results in numbers and clarify their statistical significance. So, please avoid using ‘higher’. Are the results statistically significant? This subsection should close with one to two sentences which put the results into a more general context. The conclusion should include one sentence describing the main result using such words like “Here we show”. The conclusion should write the potential and the advance this study has provided in the field and 
finally a broader perspective (two to three sentences) readily comprehensible to a scientist in any discipline.

Response: We acknowledge these comments, however to our opinion, all 
these instructions are not possible within a word limit of 200 words. We have tried our best to keep the word limit of 200 words and have followed most of the suggestions. We hope that the Reviewer will be satisfied by the revised version of our abstract.

Comment #4: Keywords: Please focus on choosing ten keywords from Medical 
Subject Headings (MeSH) (https://meshb.nlm.nih.gov/). I recommend adding ‘mania’ or equivalent as a keyword and use as many as possible in the title and in the first two sentences of the abstract.

Response: We acknowledge this comment and have revised our keywords 
accordingly.

Comment #5: A graphical abstract that will visually summarize the main findings of the manuscript is highly recommended.

Response: We acknowledge this comment and now additionally provide a 
graphical abstract.

Comment #6: Introduction: The author needs to fully expand this section with up to 1000 words, as it is short of information on the main constructs of this study, which should be understood to a reader in any discipline and make persuasive enough to put forward the main purpose of current research the author has conducted and the specific purpose the authors has intended by this study. I would like to encourage the author to present the introduction starting with the general background, proceeding to the specific background, and finally the current issue addressed to this study, leading to the objectives. Those main structures should be organized in a logical and cohesive manner. In this regard, I would suggest providing a general overview of pathogenesis and biochemical hallmarks of inflammation and mental illness, for example Mitochondrial impairment as a common motif in neuropsychiatric presentation (https://doi.org/10.3390/cells11162607) and ‘Immune influencers in 
action in inflammation and kynurenines. Moreover, I would recommend adding more information on neural substrates of cognitive dysfunction presented in mental illnesses as in ‘Functional interplay between central and autonomic nervous system’. This information may provide a better understanding of prefrontal cortex’s key role and how its disrupted function may contribute to irregular behavioral responses and therefore to the development of many cognitive dysfunctions that are common in encephalopathy (https://doi.org/10.3389/fnbeh.2022.998714).

Response: We have followed the recommendation to expand our introduction
(approx. +300 words) , in order to offer a more solid scientific background for the reader (we also kindly refer to Comment #3 of Reviewer 1). However, we have not used the suggested references, as they did not suit well with the text’s flow.

Comment #7: Methods: I recommend including short introductory paragraphs 
regarding the study design and citing more references to ensure the reliability and the integrity of evidence in the study design the authors have built and the methodology the authors applied to this study. Also, I recommend using more paragraphs or subheadings according to the type of analysis, as in the results.

Response: We acknowledge these remarks and have followed the suggestions of the Reviewer.

Comment #8: Results: I would like the authors to organize this section according to the methods section, so that a reader could easily follow the results. I also expect the authors to close this section with a paragraph which put the results into a more general context.

Response: We acknowledge these remarks and have followed the suggestions of the Reviewer.

Comment #9: Discussion: The authors need to fully expand this section with up to 1500 words. Some essential elements for discussion are described, but I recommend reorganizing clarifying the following elements. Starting with the summary of the previous section (Results), the authors need to develop discussion on the potential of this study complementing as the extension of the previous work, the implication of the findings of this study, how this study could facilitate future research, the ultimate goal, the challenge, the knowledge and the technology necessary to achieve this goal, the statement about this field in general, and finally the importance of this line of research. It is particularly important to present its limit and its merit, and its potential translation of this study to clinical application.

Response: We acknowledge these remarks and have tried to follow the 
suggestions of the Reviewer. The discussion section now includes additional text 
(approx. +320 words) and briefly comments on the major issues commented on by the Reviewer.

Comment #10: Conclusion: I think that this section would benefit from a single 
paragraph (which should be shorter than the current conclusion) presenting some thoughtful as well as in-depth considerations by the author as an expert to convey the take-home message, as it is very descriptive but not enough theoretical as a conclusion should be. The author should make his effort to explain the theoretical implication as well as the translational application of their research.

Response: We acknowledge this comment and have tried to follow these 
suggestions and now provide a more theoretical and concise conclusion section with translational implications.

Comment #11: References: Please follow the guidelines of the journal 
(https://www.mdpi.com/journal/brainsci/instructions) and cite more references to support introductory information and arguments for the discussion section. Typically, the original study like this needs 60-70 references.

Response: We acknowledge these remarks and have followed the 
suggestions of the Reviewer. Our paper now lists 58 (+20) references.

Reviewer 3 Report

This is an interesting cross-sectional study on immune cell ratios as diagnostic biomarkers in affective disorders. The paper is well-written and I agree that the finding may contribute well to the literature. I have several comments to improve the manuscript further:

1. First, given that the current study is a retrospective, cross sectional study, the title "Immune cell ratio as diagnostic biomarkers in affective disorders" seems inappropriate and should be further toned down. The title may give an inaccurate impression that immune cell ratios is already an established diagnostic biomarkers in affective disorder.

2. It will be helpful for the authors to supply more information on how the participants were recruited in the current study.

3. The authors should also supplement more information on how the blood sample was collected and analyzed (e.g., measurements, inter-assay and intra-assay coefficients of variance, data cleaning procedure, etc).

4. There is a need to clarify whether the analysis has corrected for multiple comparison due to the post-hoc nature. This is an important consideration when evaluating the validity of the study.

Author Response

Comment #1: This is an interesting cross-sectional study on immune cell ratios as diagnostic biomarkers in affective disorders. The paper is well-written and I agree that the finding may contribute well to the literature.

Response: We appreciate these positive comments!

Comment #2: First, given that the current study is a retrospective, cross sectional study, the title "Immune cell ratio as diagnostic biomarkers in affective disorders" seems inappropriate and should be further toned down. The title may give an inaccurate impression that immune cell ratios is already an established diagnostic biomarkers in affective disorder.

Response: We acknowledge this comment and have revised our title to be 
more descriptive and unbiased. We also kindly refer to Comment #2 of Reviewer 2.

Comment #3: It will be helpful for the authors to supply more information on how the participants were recruited in the current study.

Response: We would like to thank the Reviewer for this comment. However, 
we are not really sure on what additional information is requested by the Reviewer, given that this was a retrospective study, based on standardized inclusion/exclusion criteria described in detail in our methods section.

Comment #4: The authors should also supplement more information on how the blood sample was collected and analyzed (e.g., measurements, inter-assay and intra-assay coefficients of variance, etc).

Response: We would like to thank the Reviewer for pointing out this issue. We 
acknowledge this comment and now provide a detailed description of blood sampling and assessment in our methods section.

Comment #5: There is a need to clarify whether the analysis has corrected for multiple comparison due to the post-hoc nature. This is an important consideration when evaluating the validity of the study.

Response: Adjusted p values were calculated by using the Bonferroni 
correction. This was already stated in the legend of Table 1, which we have now 
highlighted

Round 2

Reviewer 1 Report

The paper is improved throughout the revision process.

Author Response

We would like to thank the Reviewer once again for all the comments that significantly improved our manuscript and are glad that the Reviewer is satisfied with our revised manuscript.

Reviewer 2 Report

3 March 2023 

Manuscript ID: brainsci-2253706

Type: Article

Title: ‘Immune cell ratios as diagnostic biomarkers in affective disorders: A retrospective, cross-sectional study’ by Koureta A et al., submitted to Brain Sciences 

Dear Authors,

I am pleased to see that the authors took my comments seriously and have solved most issues I raised in the previous round of the peer-review session. Currently, the manuscript is a well written and nicely presented research article investigating the immune cell ratios of patients with mania, bipolar depression, and unipolar depression. I just leave some comments here, which I believe help the authors improve the adequacy, the readability, and thus the quality of the manuscript to end my part of the peer-review session.

Comments:

1.      Abstract: The quality of this section has substantially improved. Nevertheless, I would like the authors to make another effort to polish this section. Please remove the subheadings according to the journal’s guidelines and focus on the conclusion: The conclusion should open with one sentence describing the main result using such words like “Here we show”. The conclusion should write the potential and the advance this study has provided in the field and finally a broader perspective (two to three sentences) readily comprehensible to a scientist in any discipline. I personally accept an abstract of 200-220 words, max 250 words.

2.      Introduction: This section has improved substantially. Nevertheless,  I would recommend adding more information on inflammation and oxidative stress which play a major role in the pathogenesis of psychiatric disorders (https://doi.org/10.3390/biomedicines9070734) and neural substrates of cognitive dysfunction presented in mental illnesses (https://doi.org/10.3389/fnbeh.2022.998714).

3.      Discussion: I recommend separating the first paragraph into two.

4.      Figures: Are there Figure 2 and Figure 3?

5.      References: Please follow the guidelines of the journal (https://www.mdpi.com/journal/brainsci/instructions). References for articles should look as follows: Beal, M.F.; Matson, W.R.; Storey, E.; Milbury, P.; Ryan, E.A.; Ogawa, T.; Bird, E.D. Kynurenic acid concentrations are reduced in Huntington’s disease cerebral cortex. J. Neurol. Sci. 1992108, 80–87.

I believe our careful work is essential to present high quality papers which assistant editors technically guarantee.

Overall, the manuscript contains four figures, three table, and 58 references. I believe that the manuscript carries important value in studying the immune cell ratios of patients with mood disorders.  I am looking forward to seeing more papers written by the same authors in future.

Thank you.

Best regards,

Reviewer

Author Response

Responses to Comments of Reviewer 2

Comment #1: I am pleased to see that the authors took my comments seriously and have solved most issues I raised in the previous round of the peer-review session. Currently, the manuscript is a well written and nicely presented research article investigating the immune cell ratios of patients with mania, bipolar depression, and unipolar depression. I believe that the manuscript carries important value in studying the immune cell ratios of patients with mood disorders. I am looking forward to seeing more papers written by the same authors in future.

Response: We appreciate these positive comments and are glad that the Reviewer is satisfied with our revisions!

Comment #2:  Abstract: The quality of this section has substantially improved. Nevertheless, I would like the authors to make another effort to polish this section. Please remove the subheadings according to the journal’s guidelines and focus on the conclusion: The conclusion should open with one sentence describing the main result using such words like “Here we show”. The conclusion should write the potential and the advance this study has provided in the field and finally a broader perspective (two to three sentences) readily comprehensible to a scientist in any discipline. I personally accept an abstract of 200-220 words, max 250 words.

Response: We have followed the Reviewer´s suggestions. The abstract now counts 240 words.

Comment #3:  Introduction: This section has improved substantially. Nevertheless, I would recommend adding more information on inflammation and oxidative stress which play a major role in the pathogenesis of psychiatric disorders (https://doi.org/10.3390/biomedicines9070734) and neural substrates of cognitive dysfunction presented in mental illnesses (https://doi.org/10.3389/fnbeh.2022.998714).

Response: We acknowledge the Reviewer´s comment. In our first revision, we have added almost 2,5 paragraphs on inflammation and immune dysregulation to the original text. We feel that this already provides a very good introductory basis to the reader and have not expanded more. However, we do now include an additional paragraph on the role of oxidative stress in inflammation according to the Reviewer´s suggestions and have included the first reference suggested.

Comment #4:  Discussion: I recommend separating the first paragraph into two.

Response: We have followed the Reviewer´s suggestions.

Comment #5:  References: Please follow the guidelines of the journal (https://www.mdpi.com/journal/brainsci/instructions). References for articles should look as follows: Beal, M.F.; Matson, W.R.; Storey, E.; Milbury, P.; Ryan, E.A.; Ogawa, T.; Bird, E.D. Kynurenic acid concentrations are reduced in Huntington’s disease cerebral cortex. J. Neurol. Sci. 1992, 108, 80–87.

Response: We have followed the Reviewer´s suggestions using the MDPI EndNote style provided by the journal (not highlighted in the text).

Reviewer 3 Report

The authors have sufficiently addressed all my comments.

Author Response

(The authors gave the same response as above.)
